# The Effects of Local Weed Species on Arbuscular Mycorrhizal Fungal Communities in an Organic Winter Wheat (*Triticum durum* L.) Field in Lebanon

**DOI:** 10.3390/microorganisms12010075

**Published:** 2023-12-30

**Authors:** Soukayna Hayek, Camille Marchal, Stéphanie Huc, Ludivine Lapébie, Sylvain Abdulhak, Jérémie Van Es, Viviane Barbreau, Bello Mouhamadou, Marie-Noëlle Binet

**Affiliations:** 1Faculty of Natural and Applied Sciences, Notre Dame University-Louaize, Zouk Mosbeh, P.O. Box 72, Zouk Mikael 5425, Lebanon; soukaynah18@gmail.com; 2Laboratoire d’Écologie Alpine, CNRS UMR 5553, Université Grenoble Alpes, CS 40700, CEDEX 09, 38058 Grenoble, France; marchal.cam38@gmail.com (C.M.); marie-noelle.binet@univ-grenoble-alpes.fr (M.-N.B.); 3Conservatoire Botanique National Alpin, Domaine de Charance, 05000 Gap, France; s.huc@cbn-alpin.fr (S.H.); l.lapebie@cbn-alpin.fr (L.L.); s.abdulhak@cbn-alpin.fr (S.A.); j.van-es@cbn-alpin.fr (J.V.E.); 4Collège Henri Wallon, 17 Rue Henri Wallon, 38400 Saint Martin d’Hères, France; viviane.barbreau@orange.fr; 5Maison de l’Université Esplanade Erasme, Université de Bourgogne, CEDEX BP27877, 21078 Dijon, France

**Keywords:** arbuscular mycorrhizal fungi, *Triticum durum*, local weed species, organic management

## Abstract

We examined the potential effects of weed species on the arbuscular mycorrhizal fungi (AMF) in an organic winter wheat (*Triticum durum*) field in Lebanon. In this agroecosystem, the field and its surroundings were covered with spontaneous vegetation corresponding to local weeds. The coexistence between wheat and weeds did not modify AM fungal community diversity and colonization in *T. durum* but changed their composition. We evidenced 22 operational taxonomic units (OTUs) specifically shared between wheat associated with weeds (Td_w_) and weeds, regardless of their localization and 12 OTUs with an abundance of variation between wheat without neighboring weeds (Td) and Td_w_. The number of AM propagules and total C and N contents were higher in soil covered with wheat associated with weeds (TdW_soil_) vs. wheat without neighboring weeds (Td_soil_). In greenhouse experiments, the shoot biomass and root mycorrhizal intensity of *Medicago sativa,* used as a trap plant, were higher using TdW_soil_ vs. Td_soil_ as the inoculum. Positive correlations were observed between soil AM propagule numbers and *M. sativa* shoot biomass, on the one hand and *M. sativa* mycorrhizal intensity, on the other hand. Weeds seemed to exert significant effects on root AM fungal composition in *T. durum* and these effects may contribute to enhanced AMF development in the field.

## 1. Introduction

The current agroecological approach attributes an important role to biodiversity conservation in agriculture fields in maintaining stable and sustainable crop production [1]. This biodiversity may strengthen the resilience of cropping systems facing multiple global changes [2]. 

Segetal plant species, commonly known as weeds, are spontaneous vegetation present in all types of field crops [3]. The Mesopotamian Region, and more precisely, the middle Eastern steppes (Jordan, Iraq, Iran, Turkey, Lebanon), is often recognized as the origin of the Western European segetal plants that have been coevolving with winter cereals since the emergence of agriculture [3,4,5]. Weeds have often been considered as undesirable species in agricultural fields by competing with crop plants for available nutrients and water, a competition sometimes in favor of weeds [6]. Agricultural technique modernization (massive uses of pesticides, deep plowing and seed cleaning techniques) has highly affected both the abundance and richness of weed flora in cereal fields. Consequently, an important decline in weed communities was revealed in many European countries [7]. 

However, more recent studies have shown that weeds can offer numerous agroecosystem services beneficial to crops [8,9,10]. This led to the development of a conservation strategy in sustainable agriculture both to stop the decline of weed plant diversity [11] and to exploit their potential in the functioning of agroecosystems. Indeed, weeds can provide ground cover that can help reduce soil erosion and N loss [12], much like a winter cover crop. They can support insect natural enemies [13], pollinators [14] and birds [15] and increase biodiversity at the field and landscape level [16]. Furthermore, an increasing number of studies have investigated the interactions between weeds and arbuscular mycorrhizal fungi (AMF) in crop systems and their importance in agroecosystem [17]. 

Arbuscular mycorrhizal fungi form symbiotic associations with most terrestrial plants [18]. As obligate mutualistic symbionts, AMF acquire water and key nutrients, including phosphorus (P) and nitrogen (N), through the extraradical mycelium acting as an extension of the host root system, and transfer them to the host plant in exchange for photosynthetically assimilated carbon (4 to 20% of total fixed carbon) [19]. They provide a range of Important ecological services, in particular by improving plant nutrition, stress resistance and tolerance, soil structure and fertility. AMF also interact with most crop plants including cereals, vegetables and fruit trees; therefore, they receive increasing attention for their potential use in sustainable agriculture [20]. The structure and composition of AM fungal communities are influenced by a combination of various factors and complex interactions. Agricultural practices such as fertilizer and pesticide applications, soil tillage and diverse cropping systems are common abiotic factors that may substantially alter AM fungal communities [21]. Biotic factors including plant species were also reported to strongly influence the AM fungal community, likely through signaling compounds and species-specific traits [22,23]. 

While numerous studies showed that weeds can promote AMF colonization and AM proliferation in soil in controlled systems [24,25], their potential effects on the diversity and the composition of AM fungal communities in crop fields are not well known [26]. It is therefore important to accentuate studies in agroecosystems to understand the impact of weeds on AMFs and their consequences on crops.

The main purpose of this article was to study the effect of weeds on AM fungal community diversity and composition in a winter wheat field in the south of Lebanon. In this agroecosystem, the management practice used was organic farming, i.e., farming without the use of synthetic chemicals or fertilizers and including crop rotations and reduced tillage. The field and its surroundings were covered with spontaneous vegetation that corresponded to local weed species. Using real field conditions (uncontrolled field associations between neighboring weeds and wheat plants), we characterized, via high-throughput sequencing, the fungal communities associated with the roots of weeds and wheat in plots containing each and both types of plant communities. We also investigated the effects of weeds on the functional performance of soil by characterizing the abundance of soil AMF propagules and by studying the soil’s capacity to enhance the biomass and root AMF colonization of a host plant *Medicago sativa* in pot culture experiments.

We tested the hypothesis that the presence of weeds would modify AM fungal community colonization, diversity and composition in the roots of wheat crops and these modifications might have positive effects on the functional performance of soil in terms of AMF propagule abundance, growth and root colonization in the host plant. 

## 2. Material and Methods

### 2.1. Site Description 

The study site is a small-scale farming system of one winter wheat landrace species in the south of Lebanon (Hanaway, 33°13′16″ N 35°16′39″ E; annual mean temperature from 2018 to 2022: 25.52 ± 0.1 °C; annual mean humidity from 2018 to 2022: 72.83 ± 2.21 °C). The soil is fairly deep, and has a clay-loam texture corresponding to CL code according to FAO classification with an important water-holding capacity. The soil was exploited to cultivate wheat in the third consecutive year, without irrigation and without chemical input. Sowing was carried out on 2 December 2021, in rows separated by 25 cm at a density of 210 Kg of seeds per hectare. Local weed species covered about 10–15% of the surface of the wheat field. The edge outside of the field was covered with local weeds and stones covering about 70% and 20% of the surface, respectively. In our study, we selected one area of 20 m × 20 m inside the wheat field and another one of 20 m × 3 m on the edge outside of the wheat field. 

### 2.2. Experimental Approach 

Soil and root samples were collected in the studied areas in May 2022, at the ripening stage of wheat (BBCH 89). Inside the field area, we randomly collected 5 soil block samples of a 20 cm diameter and 15 cm depth covered only with *T. durum* (3 plants) due the natural absence of weeds and 5 other soil blocks with *T. durum* naturally surrounded by weeds. We also collected 5 soil block samples covered with weeds from the edge outside the field. In addition, at each location of the soil block samples, we performed floristic inventory in 0.5 × 0.5 m quadrats [27]. All of the weed species enumerated in the quadrat were identified (Appendix A) (plant records from Conservatoire Botanique National Alpin, Gap, France). The collected soil samples from the soil blocks corresponded to Td_soil_ (soil from block samples covered with *T. durum* without neighboring weed plants), TdW_soil_ (soil from block samples covered with *T. durum* surrounded with weeds) and W_soil_ (soil from block samples covered with weeds in the border outside of the wheat field). The resulting 15 soil samples (5 biological replicates per condition Td_soil_, TdW_soil_, W_soil_) were passed through 5 mm-sieves and used for chemical analysis, the quantification of mycorrhizal intensity and plant productivity experiments. The roots from each soil block sample were carefully collected and thoroughly rinsed with tap water, excised in 2–5 mm fragments and kept in the freezer at −80 °C until further processing. In the case of soil blocks covered with *T. durum* and weeds, the roots of *T. durum* were separated from those of weed species roots to distinguish Td_w_ (*T. durum* surrounded with weeds) and W_Td_ (weeds associated to *T. durum*) root samples. The other types of root samples were Td (*T. durum* without neighboring weeds) and W (weeds located on the edge outside the field) isolated, respectively, from Td_soil_ (soil samples covered with *T. durum* without neighboring weed plants) and W_soil_ (soil samples covered with weeds on the edge outside of the field). The resulting 20 root samples (5 biological replicates per condition Td, Td_w_, W_Td_ and W) were used for the determination of the level of mycorrhizal colonization and DNA extraction. 

Plant tissues were collected in accordance with relevant institutional, national and international guidelines and legislation. Permits to export samples and to conduct the research were granted by the Union of Tyre Municipalities (Lebanon) and the Université Grenoble Alpes (France).

### 2.3. Soil Chemical Properties 

Total carbon (%), total nitrogen (%) and pH were measured from each sampled soil by Aurea, France according to standard methods. The soil pH in the three studied soil samples (Td_soil_, TdW_soil_ and W_soil_), did not change, with an average of 8.10. However, the soil total C and N contents were significantly higher in TdW_soil_ in comparison to Td (1.5 and 1.3 times respectively) (Appendix A). There were no significant differences in soil total C and N contents between W and Td samples (Appendix A). 

### 2.4. The Determination of Weed Species Richness

Weed species richness corresponded to the total number of different weed species in each studied condition.

### 2.5. The Determination of the Density and Viability of AMF Propagules in Soils

The total number of AMF propagules was estimated using a soil-dilution method, the “most probable number” (MPN) method to assess propagule viability [28,29]. These propagules corresponded to different infective AMF forms: spores, external and internal hyphae (colonized root fragments).

### 2.6. Target Plant Growth Experiments in Greenhouse

To assess whether weeds can influence AMF’s effect on plant growth and root colonization, one highly mycorrhizal species, *M. sativa* was used as a host trap plant because it grows well under greenhouse conditions and is a suitable host for a wide variety of AM fungi [29]. Seeds of *M. sativa* were germinated on moist filter paper, and 2-day-old seedlings were transplanted individually into pots containing a 150 mL mix of field soil (100 mL) and sterilized sand (50 mL) and grown under greenhouse conditions. 

For each condition, Td_soil_, TdW_soil_ and W_soil_, we used 15 replicates of the mix of field and sand soil, giving a total of 45 pots. One *M. sativa* plantlet was transplanted into each pot. Each week, the 45 pots received 20 mL of Long Ashton nutrient solution [30] containing 13.4 μM of phosphate (provided as NaH_2_PO_4_ 2H_2_O), 8 mM of nitrate (provided as KNO_3_ and Ca(NO_3_)_2_ 2H_2_O) and 70 μM of ammonium (provided as (NH_4_)_6_Mo_7_O_24_ 4H_2_O). After 6 weeks, the *M. sativa* plants were analyzed for mycorrhizal intensity and aerial plant biomass. 

### 2.7. The Level of Mycorrhizal Root Colonization

Roots were washed to remove soil particles and partially digested in 10% KOH (*w*/*v*) at 90 °C for 1 h. After digestion, the roots were stained with 0.05% trypan blue in lactophenol (*v*/*v*) at 90 °C for 30 min [31]. Fungal structures (hyphae, vesicles, arbuscules) were stained in blue. AMF colonization was estimated under light microscopy from 30 root fragments of approximately 1 cm in length [32]. Mycorrhizal intensity was evaluated on the MYCOCALC program (http://www.dijon.inra.fr/mychintec/Mycocalc-prg/download.htm (2001)) (accessed on 27 September 2023). and was an estimation of the amount of root cortex infected by AMF.

### 2.8. DNA Extraction, PCR Amplification and Illumina-Based Sequencing 

The 20 root samples were dried and ground with liquid nitrogen and 100 mg were used to extract DNA with the Fast prep DNA Spin kit (MPBIO, Illkirch-Graffenstaden, France) according to the manufacturer’s instructions. The AMF-specific universal primers FLR3 and FLR4 [33] were used to amplify the partial large subunit region. The primers were extended using sample-specific tags of 8 nt length to allow the parallel sequencing of multiple samples. The PCR reactions were carried out as reported [22], adjusting the annealing temperature (58 °C) according to the FLR3 and FLR4 primers. 

Four PCR reactions (technical replicates) were carried out using the DNA extracted from each sample and the four PCR products from each sample were pooled. PCR products were purified with the QIAquick kit in accordance with the manufacturer’s instructions (Qiagen, Courtaboeuf, France) and DNA was quantified using the Bioanalyzer (Agilent Technologies, Inc., Santa Clara, CA, USA). The purified amplicons were pooled for the sequencing by using equivalent molarities amongst amplicons. The library construction and sequencing (Illumina MiSeq 150 bp pair-end) were carried out by Fasteris (Geneve, Switzerland).

### 2.9. Bioinformatic Analysis

Reads’ assembly and primary filtering were performed using the OBItools package [34]. The ecotag function of OBItools was used to taxonomically assign the obtained unique sequences (identical repeated sequences) to the r143 EMBL fungi database using the FLR3-FLR4 fungi primers. Short (<100 nt) or rare (occurrence < 2) reads were removed. Unique sequences belonging to the fungal Kingdom were selected, resulting in 39,701 unique sequences for 1,119,249 reads in total. Highly similar unique sequences were clustered into OTUs by computing pairwise similarities with Sumatra (OBItools package, https://metabarcoding.org/sumatra, accessed on 11 April 2023) and forming 97% similarity clusters with the Markov Cluster algorithm (MCL) classification process [35]. 

The MetabaR package [36] was used on the R software 4.2.3 to flag and filter out spurious signals. The extraction, PCR and sequencing of control samples allowed for flagging of contaminants (0.12% of OTUs, 0.011% of reads). Non-target taxa were detected based on a minimum best identity score of 0.8 (0.13% of OTUS, 0.017% of reads). Tag-jumps were detected and removed using a minimum abundance threshold of 10^−2^ (0.02% of OTUs, 0.77% of reads). Finally, PCR outliers were flagged if the compositional dissimilarities in OTUs between PCR replicates from the same sample were not lower than that observed between PCRs obtained from different samples, as well as based on a minimum sequencing depth of 10^3^ reads per PCR (4.4% of PCRs, 8.1% of reads). Samples with a low OTU count were removed (1 sample, 1.18% of reads).

The dataset was normalized by sample using the Hellinger standardization method (“descostand” function, package vegan) [37]. Unique sequences belonging to the same OTUs were aggregated. The final contingency table displays the relative abundance (normalized read counts) of 506 OTUs among 19 samples.

### 2.10. Statistical Analysis 

The normality of the data was tested by the Shapiro–Wilk test and homoscedasticity was tested using Levene’s test. Means were compared with ANOVAs followed by the Tukey’s Honest Significant Differences test, or the Kruskal–Wallis rank sum test for non-parametric data. The fungal alpha diversity indexes were calculated using the vegan package and compared with ANOVAs followed by the Tukey’s HSD test, or the Kruskal–Wallis rank sum test for non-parametric data. The distribution of fungal communities was evaluated using Principal Component Analysis (PCA) based on the sequences of OTUs present in the four conditions. Soil chemical parameters, weed species richness and mycorrhizal intensity for *T. durum* and weeds species are displayed as supplementary quantitative variables to the PCA. Pairwise comparisons between modalities were performed using PERMANOVA on Bray–Curtis distance matrices (adonis2 function), followed by a Bonferroni correction. The variance partitioning of beta diversity was measured using PERMANOVA on Bray–Curtis distance matrices, followed by the Tukey Honest Significant Differences test. Linear regressions were calculated using Pearson’s coefficient. All statistical analyses were performed using the R package vegan [37].

### 2.11. Data Availability 

All of the fungal sequences used in this study are openly available in the Genbank database at the following link: https://www.ncbi.nlm.nih.gov/sra/?term=SRR26165734 (accessed on 27 September 2023). Among these sequences, the fungal sequences used in molecular phylogeny were also submitted separately to the Genbank database under the accession number: OR537577-OR537629.

## 3. Results 

### 3.1. The AMF Colonization and Alpha Diversity of Root AM Fungal Community

The mycorrhizal intensity did not differ between the roots of the four types of root samples (Td, Td_w_, W_Td_ and W) (Appendix A).

Concerning the alpha diversity, first, rarefaction analysis performed on the data of each of the four studied samples (Td, Td_w_, W_Td_ and W) indicated that the taxon accumulation curves reached saturation (Appendix A). The richness and the evenness of AM fungal communities in the roots of *T. durum* and weeds did not significantly vary between the four types of studied samples (Appendix A). 

### 3.2. AM Fungal Community Distribution

Principal components analysis (PCA) based on the sequences of all OTUs, followed by pairwise comparison tests identified distinct AM fungal groups with significant differences between W and Td (*p* = 0.008) as well as between W and Td_w_ (*p* = 0.009). Similar to the pattern observed in the alpha diversity, important variabilities in the AM fungal communities were observed in the Td_w_, W_Td_ and W samples compared to those of Td samples (Figure 1). The weed species richness and the soil total nitrogen were the significant factors determining the AM fungal beta diversity, explaining, respectively, 19% (*p* = 0.001) and 11% (*p* = 0.021) of the variance (Table 1). The other measured parameters (pH and soil total C) did not significantly influence the AM fungal beta diversity (Table 1).

### 3.3. AM Fungal Community Composition 

From Venn analysis based on all OTUs, we observed that 146 OTUs were shared between all samples and represented at least 92% of the total abundance (Table 2). We were interested by the quantitative and qualitative differences in OTUs between Td and Td_w_ roots. 

Concerning Td_w_, we identified 22 OTUs shared between Td_w_ and weeds regardless of their localization (W_Td_ and W) but not present in wheat alone (Td), OTUs that could probably be present in Td_w_ due to the presence of weeds. These 22 OTUs represented 0.84% of the total abundance in Td_w_. These 22 OTUs belonged to five genera: *Funneliformis* (7 OTUs), *Septoglomus* (3 OTUs), *Kamienskia* (1 OTUs), *Rhizophagus* (8 OTUs) and *Glomus* (2 OTUs), and one OTU, which was not identified at the genus level, belonged to Glomeromycotina (Figure 2). Concerning Td, 15 OTUs were present only in Td and weeds (regardless of their localization), even though the two types of plants were not nearby. These 15 OTUs represented 0.62% of the total abundance in Td (Table 2). 

Moreover, we evidenced 12 OTUs shared between Td_w_ and Td, that varied quantitatively (Table 3). Among them, ten OTUs (*Funelliformis* sp5, *Funelliformis* sp7, *Funelliformis* sp8, *Funelliformis* sp12, *Funelliformis* sp14, Glomeromycotina 1, *Rhizophagus* sp1, *Rhizophagus* sp2, *Rhizophagus* sp3 and *Rhizophagus* sp4) were more abundant in Td_w_ and two OTUs (*Rhizophagus* sp14 and *Glomus* sp2) were less abundant as compared to Td samples (Table 3; Figure 2).

In summary, the results showed qualitative and quantitative differences in AMF composition between Td_w_ and Td. Qualitative differences concerned 22 OTUs that were present in Td_w_ roots but absent in Td roots. As for quantitative differences, 12 OTUs showed differences in abundance between Td_w_ and Td. Among these OTUs, ten and two OTUs were more abundant in Td_w_ and Td roots, respectively.

### 3.4. AMF Propagules 

The number of soil AMF propagules was significantly higher in TdW_soil_ samples (2308 ± 429 propagules.kg^−1^ soil), at least 2.4 or 8.7 times more than in W_soil_ (950 ± 434 propagules.kg^−1^ soil; ANOVA, *p* = 0. 00803) or Td_soil_ samples (265 ± 68; ANOVA, *p* = 0.00099), respectively.

### 3.5. The Indirect Effect of Weeds on M. sativa Trap Plant Productivity and Mycorrhization in Greenhouse 

The *M. sativa* shoot dry weight (DW) biomass was significantly higher in TdW_soil_ (+189%; *p* = 0.01613) compared to that of Td_soil_ (Appendix A). Moreover, the mycorrhizal intensity of *M. sativa* was significantly higher (2.6-fold; *p* = 0.00107) in TdW_soil_ compared to that of Td_soil_ (Appendix A). Positive correlations were observed between the abundance of AMF propagules in Td_soil_ or TdW_soil_ samples and the *M. sativa* DW biomass on the one hand (*p* = 0.01867), and the *M. sativa* mycorrhizal intensity on the other hand (*p* = 0.00017) (Figure 3). 

## 4. Discussion

Considering the importance of AMF in all agroecosystems, the challenge is to optimize the benefits from AMF-crop associations in these systems by using agricultural practices favorable to AMF development. Several studies have reported greater colonization and AMF proliferation in soil under controlled agroecosystems in the presence of weeds [24,25]. However, under real field conditions, there is no information on how the presence of weeds impacts the crop root AMF community and soil AMF abundance, by considering both biotic and abiotic factors [26].

We investigated the effects of local weeds on the root AMF community in *T. durum* in an organic field in Lebanon, a region of the Mediterranean Basin that remains scarcely studied. We found significant differences in root AMF beta diversity between weeds on the edge outside of the field and wheat plants associated or not associated with weeds. These differences due to weed species richness and soil nitrogen are consistent with most of the studies showing the effects of biotic (host identity, plant traits) and abiotic factors on AMF beta diversity [38,39]. However, when weeds and wheat coexisted inside the field, no changes occurred in the root AMF beta diversity between weeds and wheat plants associated or not associated with weeds. This result contrasted with those of several studies showing that co-occurring plant species can harbor root AMF communities whose colonization and structures differ from those present in their roots when they develop separately, due to interactions among plant species [40,41]. We can suggest that the lack of difference in the AMF beta diversity between wheat and weeds inside the field may be due to the low species richness of this weed community compared to that of weeds outside of the field. This low weed richness may explain the lack of interactive effects between this weed community and wheat.

Even if the beta diversity of root AMF was not different between wheat associated or not associated with weeds, we observed differences in the OTU composition between these two conditions. These differences were reflected by the presence of (i) 22 OTUs shared between wheat associated with weeds and weeds regardless of their localization but lacking in wheat alone and (ii) 12 OTUs shared between wheat associated or not associated with weeds but varying in abundance. Such differences, whether qualitative or quantitative, can be explained in several ways. Firstly, the proximity of weed species might be invoked and might promote AMF recruitment in the roots of wheat by different pathways. It could be by direct contact between root systems or by a common mycorrhizal network interconnecting plants [42]. Interestingly, a very recent study showed that weed species were able to transmit their root mycobiota to nearby wheat roots in controlled conditions [43]. It could also be an indirect influence of weeds on AMF composition in *T. durum* roots that can be achieved through plant—plant interactions via root exudation. Indeed, root exudate signals produced in the soil by neighboring plants could favor AMF species selection and promote their recruitment by host plants [44]. Secondly, the quantitative and qualitative differences in OTUs between wheat associated or not associated with weeds could be independent of weeds since, we found, in particular, 15 OTUs shared between wheat alone and weeds outside of the field, without the two types of plants interacting. These OTUs could correspond to generalist species possessing a broad ecological niche. 

We also found that AM propagule abundance was higher in soil covered with both wheat and weeds. These results obtained under real field conditions suggested the positive influence of weeds on AMF propagules, consistent with a study reporting that weeds increased AMF proliferation in the soil in a wheat-controlled system [24]. Even if we cannot exclude the intrinsic ability of each AMF species to proliferate actively (i.e., to produce more extensive extraradical hyphae and/or spores) [45] or not in soil, the coexistence of wheat and weeds in the field could promote the abundance of propagules in soil in line with the studies demonstrating a greater development of propagules in co-culture than in mono-culture [46].

More interestingly, the high AM propagule abundance produced in the soil in the presence of weeds had positive effects on the plant biomass and AM root colonization of *M. sativa* plantlets in pot culture in a greenhouse as shown by studies demonstrating the same patterns in controlled experiments [26]. It is also possible that these propagules are one of the factors responsible for the modifications in soil physico-chemical properties resulting in higher C and N contents in soil samples from blocks covered with wheat associated with weeds, as they are constituted and/or able to produce organic molecules such as chitin or glomalin, respectively [47,48]. 

## 5. Conclusions

To our knowledge, this study is the first investigation on native AMF communities in a Lebanese agroecosystem. We demonstrated that the coexistence of weeds and wheat under real field conditions led to (i) qualitative and quantitative changes in the composition of wheat root AMF communities, (ii) an increase in soil AMF propagules and (iii) an improvement of soil performance on *M. sativa* productivity. While our study was carried out on one winter wheat variety in one agroecosystem at one wheat stage, future studies on several agroecosystems would be interesting to confirm the beneficial effects of weeds on crop systems. This could open the way towards the use of these weeds in sustainable agriculture.

## Figures and Tables

**Figure 1 microorganisms-12-00075-f001:**
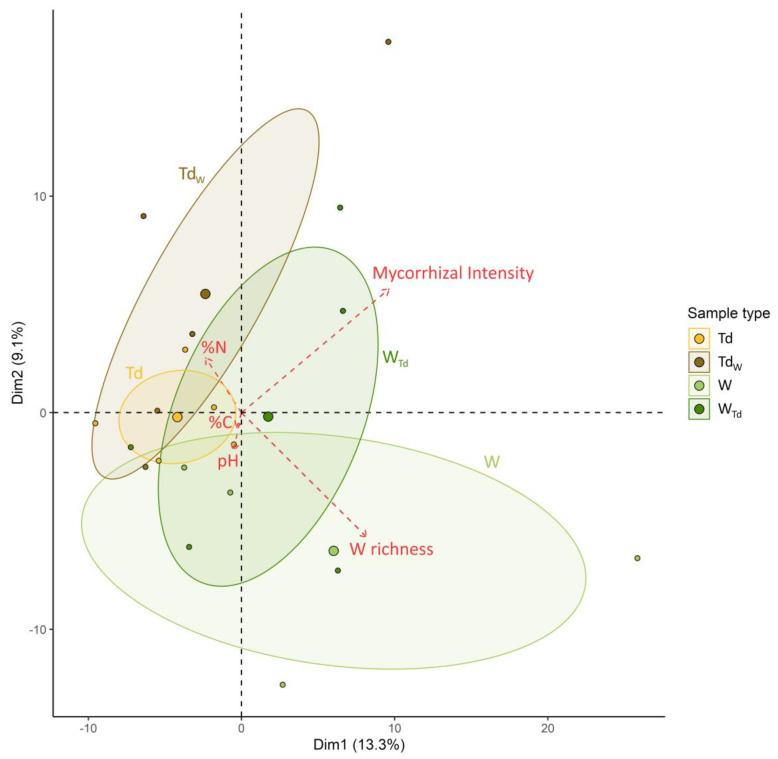
Principal components analysis of AMF communities in *T. durum* and weed root samples (Td: *T. durum* without neighboring weeds; Td_w_: *T. durum* surrounded by weeds; W_Td_: weeds associated with *T. durum*; W: weeds on the edge outside of the field). Ellipses represent 95% normal probability.

**Figure 2 microorganisms-12-00075-f002:**
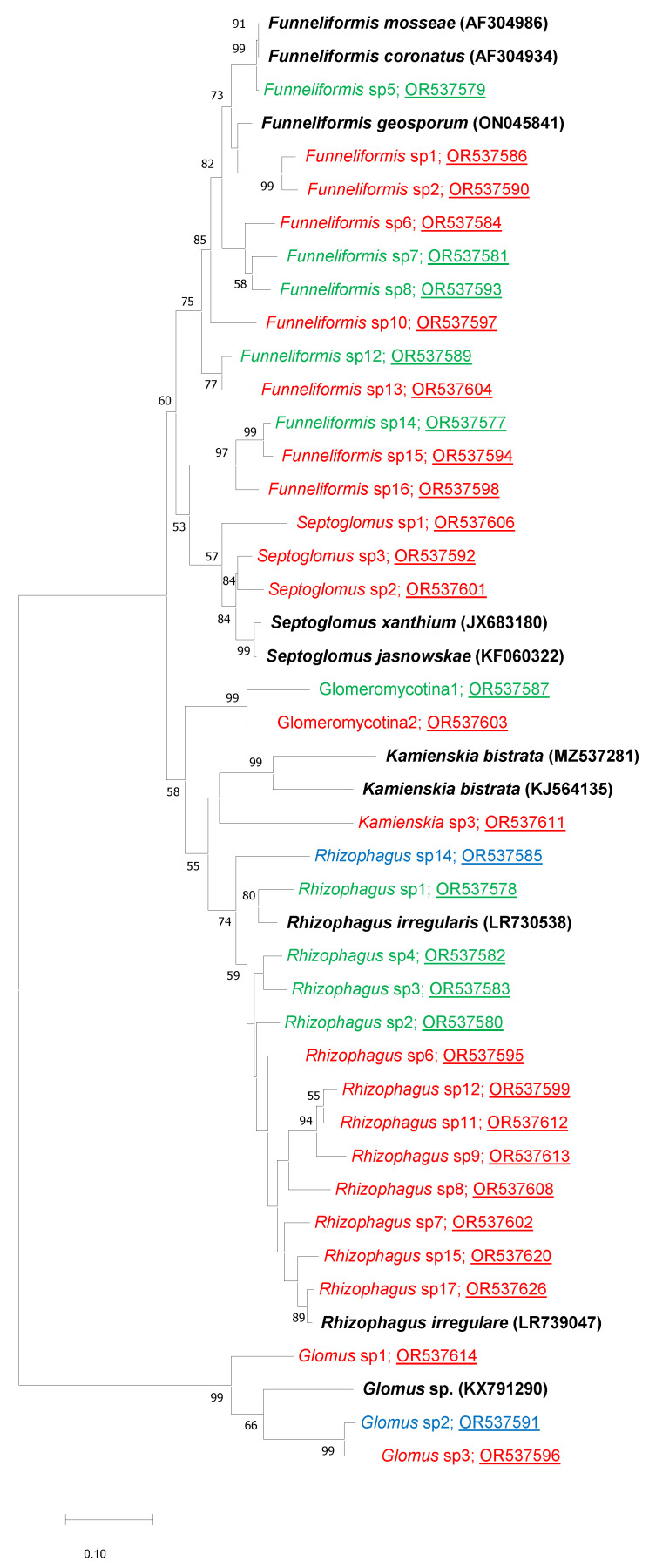
Phylogenetic analyses performed with the neighbour-joining method based on the aligned sequences of the 28S ribosomal RNA gene. Analyses were performed as described [21]. The AMF species from which the sequences have been recovered from GenBank (Genbank accession number in brackets) and used as reference sequences are indicated in black. The Genbank accession numbers from this work are underlined. Bootstrap values exceeding 50% are shown on the branches. The 22 OTUs specifically shared between Td_w,_ W and W_Td_ are indicated in red. Among the 12 OTUs showing abundance in variation between Td and Td_w_, we indicated in green the 10 OTUs that were significantly higher in Td_w_ as compared to Td and in blue the two OTUs that were less abundant in Td_w_ as compared to Td.

**Figure 3 microorganisms-12-00075-f003:**
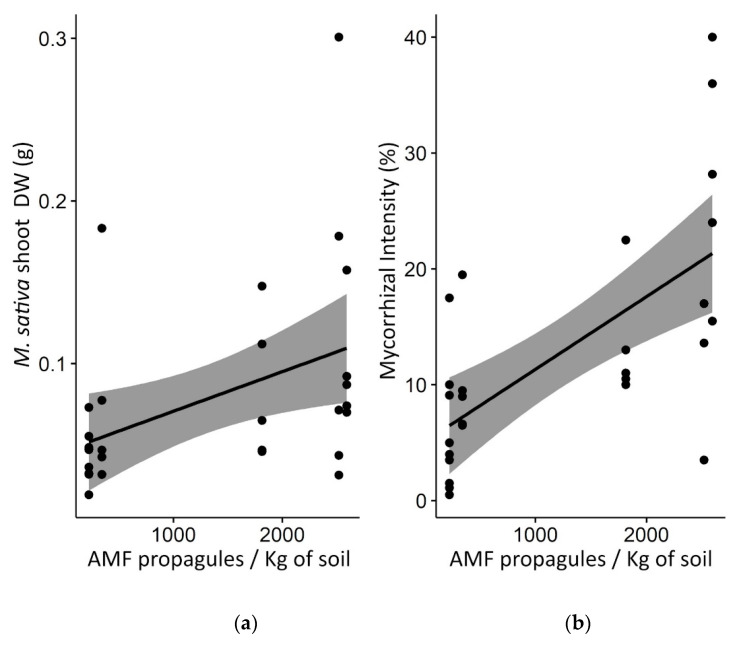
Relationship between AMF propagules and *M. sativa* biomass (**a**) or mycorrhizal intensity in *M. sativa* roots (**b**).

**Table 1 microorganisms-12-00075-t001:** Variance partitioning results of AMF communities in *T. durum* and weed root samples. (Td: *T. durum* without neighboring weeds; Td_w_: *T. durum* surrounded by weeds; W_Td_: weeds associated to *T. durum*; W: weeds on the edge outside of the field). Permutational multivariate analysis of variance (PERMANOVA) fitting a polynomial regression to a Bray-Curtis distance matrix using a permutation test with pseudo-F ratios (999 permutations).

	Df	Sum of Squares	R^2^	F	Pr (>F)
Weed species richness	1	0.20320	0.19164	4.5835	0.001
Total soil N (%)	1	0.11813	0.11141	2.6646	0.021
Total soil C (%)	1	0.04023	0.03794	0.9074	0.428
Soil pH	1	0.03374	0.03182	0.7610	0.623
Residuals	15	0.66500	0.62718		
Total	19	1.06029	1.000		

Numbers in bold correspond to significant *p*-values.

**Table 2 microorganisms-12-00075-t002:** Number and percentage of total relative abundance in OTUs in *T. durum* and weed root samples (Td: *T. durum* without neighboring weeds (*n* = 5); Td_w_: *T. durum* surrounded by weeds (*n* = 5); W_Td_: weeds associated with *T. durum* (*n* = 5); W: weeds on the edge outside of the field (*n* = 4)). The OTUs shared between all samples and the OTUs shared specifically between Td_w,_ W and W_Td_, as well as the OTUs shared specifically between Td, W and W_Td_ are indicated.

OTUs	Number of OTUs	Total Relative Abundance (%)			
		Td	Td_w_	W	W_Td_
OTUs shared between all conditions	146	94.43	92	94.23	92.78
OTUs shared between Td_w,_ W and W_Td_	22		0.84	1.02	1.66
OTUs shared between Td, W and W_Td_	15	0.62		0.72	0.83

**Table 3 microorganisms-12-00075-t003:** The relative abundance of AMF OTUs shared between *T. durum* and weed root samples (Td: *T. durum* without neighboring weeds; Td_w_: *T. durum* surrounded by weeds). Normality and homoscedasticity of data were tested using Shapiro–Wilk test and Levene’s test, respectively. One-way ANOVA tests were used to compare the average abundance between Td and Td_w_, only OTUs showing significant differences are displayed. Data represented are mean (sd), *n* = 10.

	Td	Tdw	Pr (>F)
*Funneliformis* sp5	0.020 (0.019)	0.076 (0.047)	0.040
*Funneliformis* sp7	0.020 (0.006)	0.028 (0.005)	0.041
*Funneliformis* sp8	0.001 (0.003)	0.008 (0.005)	0.039
*Funneliformis* sp12	0.001 (0.003)	0.007 (0.004)	0.045
*Funneliformis* sp14	0.015 (0.011)	0.044 (0.021)	0.030
*Glomus* sp2	0.011 (0.007)	0.002 (0.004)	0.037
*Rhizophagus* sp1	0.030 (0.010)	0.048 (0.013)	0.041
*Rhizophagus* sp2	0.025 (0.090)	0.036 (0.004)	0.040
*Rhizophagus* sp3	0.005 (0.003)	0.011 (0.003)	0.013
*Rhizophagus* sp4	0.018 (0.003)	0.024 (0.003)	0.016
*Rhizophagus* sp14	0.016 (0.008)	0.003 (0.004)	0.011
*Glomeromycotina 1*	0.006 (0.004)	0.012 (0.003)	0.026

OTUs significantly more abundant in Td_w_ and Td are indicated in red and green, respectively.

## Data Availability

Data are contained within the article and Appendix A.

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
