# Peer review of "The Effects of Local Weed Species on Arbuscular Mycorrhizal Fungal Communities in an Organic Winter Wheat (Triticum durum L.) Field in Lebanon"

_microorganisms, 2023, doi:10.3390/microorganisms12010075_

Round 1
Reviewer 1 Report
Comments and Suggestions for Authors
The manuscript entitled "Effects of local weed species on arbuscular mycorrhizal fungal communities in an organic winter wheat (Triticum durum L.) field in Lebanon” investigated the effect of weeds on AM fungal community diversity and composition in winter wheat field in the South of Lebanon. So, soil and root samples were collected form 3 sites: T. durum in the absence of weeds, T. durum naturally surrounded by weeds, and weeds from the edge outside the field. Soil chemical properties, level of mycorrhizal root colonization, and other parameters were estimated; in addition, AM fungal community colonization, diversity and composition in roots of wheat crops were evaluated.
Comments:
The existing study failed to explore the influence of weeds on both crop root arbuscular mycorrhizal fungi (AMF) community and soil AMF abundance in real field conditions, taking into account both biotic and abiotic factors. The composition and diversity of the soil AMF community were not assessed in the collected soil samples.
Anticipated was the identification of significant differences in root AMF diversity between weeds at the field's periphery and wheat plants, whether associated with weeds or not. Consequently, the question arises: What are the implications of the results obtained? While numerous explanations could be proposed, it is essential to acknowledge that without evaluating soil AMF community composition and diversity, several aspects of the data remain unaddressed.
The reported outcomes did not substantiate the mention that weeds have beneficial effects on AMF communities in a winter wheat organic field in Lebanon under authentic field conditions. Furthermore, the results contradicted previous findings mentioned in lines 370-372, suggesting that weeds play a role in enhancing AMF community development in the soil by fostering the growth of AMF propagules. These propagules could potentially serve as an AMF reservoir for future crops, aiding in the colonization of subsequent crops by AMF, a concept not supported by the present study.
Comments on the Quality of English LanguageMinor editing of English language required
Author Response
Grenoble, December 22, 2023
Dear Editor
Thank you for your consideration of the manuscript (microorganisms-2790090). We wish to thank the reviewer 1 for their reading of the manuscript. We do not agree with their stance and have justified ours in the following answers. All modifications have been incorporated in the revised text (highlighted in yellow).
Reviewer 1
Comments:
The existing study failed to explore the influence of weeds on both crop root arbuscular mycorrhizal fungi (AMF) community and soil AMF abundance in real field conditions, taking into account both biotic and abiotic factors. The composition and diversity of the soil AMF community were not assessed in the collected soil samples.
We do not agree with this remark.
First, our article is devoted to study the influence of weeds on root AMF communities in wheat field taking into account both biotic and abiotic factors.
For that:
- we determined the diversity and composition of AMF communities in the roots of wheat associated to weeds and compared them to those present in wheat roots without weeds,
- we also characterized the AMF communities in the roots of weeds alone,
- we measured several soil parameters such as C, N and pH.
To observe the benefits of the influence of weeds on wheat root AMF, we considered the AMF propagules in soils which represent different infective AMF forms: spores, external and internal hyphae (colonized root fragments). All these forms are able to initiate and spread host root colonization. We studied the AMF propagule abundances by counting their number and we examined their impact on the productivity of a Medicago sativa plant by using field soils as substrate in pot-controlled experiments.
Second, it may be important to study the diversity and composition of AMF communities in soil as suggested by the reviewer 1 or even to study other biotic parameters, we agree with that. However, we targeted the AMF communities present in roots because:
- they are selected by the plant according to their identity and that of the plant,
- they establish a close relationship with plants via mycorrhizal symbiosis.
Many relevant studies have targeted only root AMF communities and have shown their crucial role in plant diversity and productivity. For example, Higo et al., (2020a) and Garcia-Gonzalez et al., (2023) studied the impact of cover crop on the AMF based only on root AMF communities as in our case. Similarly, Higo et al., (2020b) studied the impact of P on the AMF by focusing only on the roots AMF communities. In addition to investigating the diversity and composition of AMF in roots, the advantage of our work is to quantify soil AMF by determining the number of propagules which represent the active and infectious forms of AMF.
- Higo, M., Tatewaki, Y., Iida, K., Yokota, K., & Isobe, K. (2020a). Amplicon sequencing analysis of arbuscular mycorrhizal fungal communities colonizing maize roots in different cover cropping and tillage systems. Scientific Reports, 10(1), 6039.
- Higo, M., Azuma, M., Kamiyoshihara, Y., Kanda, A., Tatewaki, Y., & Isobe, K. (2020b). Impact of phosphorus fertilization on tomato growth and arbuscular mycorrhizal fungal communities. Microorganisms, 8(2), 178.
- García‐González, I., Martínez‐García, L. B., Barel, J. M., Martens, H., Snoek, L. B., Hontoria, C., & Deyn, G. B. D. (2023). Cover crop identity determines root fungal community and arbuscular mycorrhiza colonization in following main crops. European Journal of Soil Science, e13427.
Anticipated was the identification of significant differences in root AMF diversity between weeds at the field's periphery and wheat plants, whether associated with weeds or not.
Indeed, the hypothesis of having significant differences in root AMF diversity between weeds on the field's periphery and wheat plants is relevant, but it remains a hypothesis that needs to be demonstrated; what we've done in this study. In addition, the study of AMF composition in weeds on the field's periphery is very important as it enables us to characterize the AMF species present in wheat roots which are potentially derived from weeds.
Consequently, the question arises: What are the implications of the results obtained? While numerous explanations could be proposed, it is essential to acknowledge that without evaluating soil AMF community composition and diversity, several aspects of the data remain unaddressed.
As for the implications of the results obtained, as reviewer 1 points out, "many explanations can be proposed". Furthermore, as indicated above, by targeting root AMF communities, we are providing important answers concerning the influence of weeds on the diversity and composition of root AMF communities, just as many relevant studies (indicated above) which have focused on root AMF, some of which have been published in your journal (Higo et al; 2020b).
The reported outcomes did not substantiate the mention that weeds have beneficial effects on AMF communities in a winter wheat organic field in Lebanon under authentic field conditions.
On the contrary, our results clearly show the role of weeds on:
- the AMF composition, with qualitative and quantitative differences in root OTU composition between wheat associated and not to weeds,
- the number of AMF propagules which is greater in soil covered by both weeds and wheat in comparison to soil covered only by wheat,
- the beneficial effect of soil covered by both weeds and wheat, on Medicago sativa productivity.
Furthermore, the results contradicted previous findings mentioned in lines 370-372, suggesting that weeds play a role in enhancing AMF community development in the soil by fostering the growth of AMF propagules. These propagules could potentially serve as an AMF reservoir for future crops, aiding in the colonization of subsequent crops by AMF, a concept not supported by the present study.
The role of weeds in enhancing the development of AMF in soil by promoting the growth of AMF propagules has been demonstrated since we have a higher number of infective AMF propagules in the soil covered by both weeds and wheat, indicating improved AMF development in this soil.
Furthermore, we indicated in perspective that the AMF propagules present in the soil can be maintained for several years and can act as AMF reservoirs to promote AMF colonization of new roots. But since we're talking about perspectives, the paragraph (lines 368 to 372 in the former version) has been removed for clarity.
In addition, the beneficial effects of weeds in wheat roots and soil have been clarified (lines 392-395).
We also have more thoroughly detailed the description of AMF propagules (lines 145-147).
We look forward to hearing from you soon,
Yours sincerely,
Dr Bello Mouhamadou
Reviewer 2 Report
Comments and Suggestions for Authors
The manuscript presents a very interesting study of the influence of weeds on arbuscular mycorrhizal fungal communities in wheat. AMF biodiversity in wheat and weed root samples from various surroundings was assessed and analyzed. In addition, the positive correlation of number of soil AMF propagules and M. sativa biomass weight was revealed. The manuscript is well written, methods are adequately described and conclusions are supported by the results obtained in the study. It can be published after some minor corrections:
1. Please, check the quality of the figures in the manuscript. Both figure 1 and 2 look pixelated. Microorganisms requires 300 dpi or higher resolution for the figures.
2. I recommend adding a small paragraph with some conclusions to section “3.3. AM fungal community composition” or some diagramm for shared and differed OTUs to improve the understanding of the results of this section.
3. Subscripts are missed in L154: “(NH4)6Mo7O24, 4H2O)” and I suppose in L263: “Tdw”.
4. Please, use the dot instead of comma in formulas of hydrates: “NaH2PO4 · 2H2O” instead of “NaH2PO4, 2H2O” throughout the manuscript.
Author Response
Grenoble, December 22, 2023
Dear Editor
Thank you for your consideration of the manuscript (microorganisms-2790090). We wish to thank the reviewer 2 for their reading of the manuscript. We agree with their comments and all modifications have been incorporated in the revised text (highlighted in yellow).
Reviewer 2
The manuscript presents a very interesting study of the influence of weeds on arbuscular mycorrhizal fungal communities in wheat. AMF biodiversity in wheat and weed root samples from various surroundings was assessed and analyzed. In addition, the positive correlation of number of soil AMF propagules and M. sativa biomass weight was revealed. The manuscript is well written, methods are adequately described and conclusions are supported by the results obtained in the study. It can be published after some minor corrections:
- Please, check the quality of the figures in the manuscript. Both figure 1 and 2 look pixelated. Microorganisms requires 300 dpi or higher resolution for the figures.
In the revised version, we have provided figures 1 and 2 in high quality.
- I recommend adding a small paragraph with some conclusions to section “3.3. AM fungal community composition” or some diagramm for shared and differed OTUs to improve the understanding of the results of this section.
We added in the revised version a paragraph at the end of the section 3.3 summarizing the differences in AMF composition between the roots of wheat associated to weeds and those of wheat alone (lines 290-294).
- Subscripts are missed in L154: “(NH4)6Mo7O24, 4H2O)” and I suppose in L263: “Tdw”.
These modifications have been done (Lines 156 to 158; 266)
- Please, use the dot instead of comma in formulas of hydrates: “NaH2PO4 · 2H2O” instead of “NaH2PO4, 2H2O” throughout the manuscript.
This modification has been done (Lines 158-159).
We look forward to hearing from you soon,
Yours sincerely,
Dr Bello Mouhamadou
Reviewer 3 Report
Comments and Suggestions for Authors
Good job.
However, an assessment of wheat productivity is not enough to complete the picture. At the studied stage of maturation, it is possible to give at least a qualitative assessment of the condition of wheat plants in association with and without weeds.
Minor remarks.
Please give in the Methodology section more detailed characteristics and a method for calculating parameters such as weed species richness and mycorrhizal intensity.
Author Response
Grenoble, December 22, 2023
Dear Editor
Thank you for your consideration of the manuscript (microorganisms-2790090). We wish to thank the reviewer 3 for their reading of the manuscript. We agree with their comments and all modifications have been incorporated in the revised text (highlighted in yellow).
Reviewer 3
Comments and Suggestions for Authors
Good job.
Thanks
However, an assessment of wheat productivity is not enough to complete the picture. At the studied stage of maturation, it is possible to give at least a qualitative assessment of the condition of wheat plants in association with and without weeds.
We agree with this remark, it would have been interesting to measure the effect of weeds on wheat productivity in situ. But we chose to do it indirectly in pots (under controlled conditions) by measuring productivity of Medicago sativa because we considered it easier and more reproducible. The results clearly show a significant positive effect of weeds on M. sativa shoot biomass. In addition, we were able to directly monitor the infectivity performance of AMF via the AMF propagules in colonizing M. sativa roots.
Minor remarks.
Please give in the Methodology section more detailed characteristics and a method for calculating parameters such as weed species richness and mycorrhizal intensity.
We have added, in the Material and Methods section, a paragraph explaining the determination of weed species richness (lines 140-142).
More details for calculating mycorrhizal intensity were added in the revised version (lines 163-169).
We look forward to hearing from you soon,
Yours sincerely,
Dr Bello Mouhamadou
Reviewer 4 Report
Comments and Suggestions for Authors
When conducting crop production, weeds are a significant negative factor that affects the development of crops and yield. Weeds often suppress the growth of agricultural plants, affect the development of soil microbiota, and weed control is an important element in crop production technology. The article is devoted to studying the influence of weeds on the development of wheat, which is one of the world's main agricultural plants, ensuring food security in many countries. Therefore, the relevance of the article is beyond doubt. There are several comments and questions about the article that are aimed at improving it.
1. Since the study was carried out over one year, in paragraph 2.1 it is necessary to provide data on the dynamics of average annual temperature and humidity over the past five years. Because these factors can influence the development of the microbial community and weeds.
2. In section 2.3, the authors studied only three soil characteristics: nitrogen, carbon and acidity. Why only these indicators, and, for example, not humus?
3. In section 2.5, when conducting an experiment in a greenhouse, why did the authors not use soil from the field, but prepare soil of a different composition?
4. In section 2.5, the authors used nutritional supplements, why? The goal of the work was to study the effect of weeds on the microbiota in organic farming without fertilizers.
5. The title of Figure 1 is too detailed. Part of the title is a description and should be transferred to the text.
6. In Figure 3, the x-axis is better to give 1.0 2.0 rather than 1.000 2.000.
7. According to the conclusions of the work, the main remark is why the authors did not study wheat yield in the experiment. They studied in detail the soil microbiota and root system of wheat. But wheat is grown for grain.
Author Response
Grenoble, December 22, 2023
Dear Editor
Thank you for your consideration of the manuscript (microorganisms-2790090). We wish to thank the reviewer 4 for their reading of the manuscript. We agree with their comments and all modifications have been incorporated in the revised text (highlighted in yellow).
Reviewer 4
Comments and Suggestions for Authors
When conducting crop production, weeds are a significant negative factor that affects the development of crops and yield. Weeds often suppress the growth of agricultural plants, affect the development of soil microbiota, and weed control is an important element in crop production technology. The article is devoted to studying the influence of weeds on the development of wheat, which is one of the world's main agricultural plants, ensuring food security in many countries. Therefore, the relevance of the article is beyond doubt. There are several comments and questions about the article that are aimed at improving it.
- Since the study was carried out over one year, in paragraph 2.1 it is necessary to provide data on the dynamics of average annual temperature and humidity over the past five years. Because these factors can influence the development of the microbial community and weeds.
In the revised version, we have indicated the average annual temperature and humidity over the last 5 years prior to 2022 (lines 92-93). These values are constant over this period, suggesting their low impact on weeds and microorganism communities over this period.
- In section 2.3, the authors studied only three soil characteristics: nitrogen, carbon and acidity. Why only these indicators, and, for example, not humus?
Indeed, we have not determined all the soil characteristics. We took a global approach by measuring the total C and total N which are present in soil compounds. Along with pH, these are the predominant factors influencing fungal communities which are classically studied.
- In section 2.5, when conducting an experiment in a greenhouse, why did the authors not use soil from the field, but prepare soil of a different composition?
Our description lacked precision; we used soil from the field. We have supplemented this information in the revised version with more details in the Material/Method section (lines 153-161).
- In section 2.5, the authors used nutritional supplements, why? The goal of the work was to study the effect of weeds on the microbiota in organic farming without fertilizers.
We used a nutrient solution to avoid mineral nutrient deficiency in the plants. We used the long Ashton solution with a low phosphate concentration so as not to inhibit the establishment of AM symbiosis. This is a classic protocol used for mycorrhizal inoculation in greenhouse experiments.
- The title of Figure 1 is too detailed. Part of the title is a description and should be transferred to the text.
We agree and have shortened the title of Figure 1 by transferring part of it to the Material/Method section (lines 217-220).
- In Figure 3, the x-axis is better to give 1.0 2.0 rather than 1.000 2.000.
We have rewritten the x-axis values which correspond to 1000 and 2000 AMF propagules/kg of soil.
- According to the conclusions of the work, the main remark is why the authors did not study wheat yield in the experiment. They studied in detail the soil microbiota and root system of wheat. But wheat is grown for grain.
We agree with this comment. As mentioned by the Reviewer 3, it would have been interesting to measure the effect of weeds on wheat productivity in situ. But we chose to do it indirectly in pots (under controlled conditions) by measuring productivity in of Medicago sativa because we considered it easier and more reproducible. The results clearly show a significant effect of weeds on M. sativa biomass. In addition, we were able to directly monitor the infectivity performance of AMF via the AMF propagules in colonizing M. sativa roots.
We look forward to hearing from you soon,
Yours sincerely,
Dr Bello Mouhamadou
Round 2
Reviewer 1 Report
Comments and Suggestions for Authors
Although the authors did not agree with my comments and justified their answer by mention that “many relevant studies (indicated in author answer) which have focused on root AMF, some of which have been published in Microorganisms journal (Higo et al; 2020b).”, I will accept the manuscript.
I understand that the study of root AMF community composition is very interested, but my comment is that without evaluating soil AMF community composition and diversity, several aspects of the data remain unaddressed during that experiment.
I feel that the authors did not catch my major comment of “The existing study failed to explore the influence of weeds on both crop root arbuscular mycorrhizal fungi (AMF) community and soil AMF abundance in real field conditions, taking into account both biotic and abiotic factors. The composition and diversity of the soil AMF community were not assessed in the collected soil samples.”. So, authors did not agree with this remark and mentioned that “First, our article is devoted to study the influence of weeds on root AMF communities in wheat field taking into account both biotic and abiotic factors.” However, I understand that this study investigated root AMF community taking into account both biotic and abiotic factors. My comment is that the influence of weeds on both crop root arbuscular mycorrhizal fungi (AMF) community and soil AMF abundance in real field conditions can not be determined without evaluating soil AMF community composition.
Another major comment is “The reported outcomes did not substantiate the mention that weeds have beneficial effects on AMF communities in a winter wheat organic field in Lebanon under authentic field conditions.” I understand that the presence of weeds affected AMF communities, that is the expected results. My question is where are your results that support the beneficial effect on winter wheat organic field?
Author Response
Grenoble, December 28, 2023
Dear Editor
We would like to thank reviewer 1 for their comments. Even if we don't agree with their vision, we have made changes to clarify our objectives.
Reviewer 1
Comments and Suggestions for Authors
Although the authors did not agree with my comments and justified their answer by mention that “many relevant studies (indicated in author answer) which have focused on root AMF, some of which have been published in Microorganisms journal (Higo et al; 2020b).”, I will accept the manuscript.
I understand that the study of root AMF community composition is very interested, but my comment is that without evaluating soil AMF community composition and diversity, several aspects of the data remain unaddressed during that experiment.
I feel that the authors did not catch my major comment of “The existing study failed to explore the influence of weeds on both crop root arbuscular mycorrhizal fungi (AMF) community and soil AMF abundance in real field conditions, taking into account both biotic and abiotic factors. The composition and diversity of the soil AMF community were not assessed in the collected soil samples.”. So, authors did not agree with this remark and mentioned that “First, our article is devoted to study the influence of weeds on root AMF communities in wheat field taking into account both biotic and abiotic factors.” However, I understand that this study investigated root AMF community taking into account both biotic and abiotic factors. My comment is that the influence of weeds on both crop root arbuscular mycorrhizal fungi (AMF) community and soil AMF abundance in real field conditions can not be determined without evaluating soil AMF community composition.
The study of soil AMF composition can be quite interesting, it can help answer scientific questions but they are not the focus of this study.
Our main objective is to study the impact of weeds on the diversity and composition of root AMF. It is possible to achieve this objective without studying the soil AMF composition, as other studies have done.
On the other hand, we explored the impact of weeds on the soils by investigating the number of soil AMF propagules (active and infectious forms of AMF). Propagule development in soil can depend on the soil AMF composition, but also on other factors such as the plant species, in particular the weeds species that can promote AMF sporulation. In all cases, by comparing the number of propagules in two different soils (soil in the presence of weeds and wheat and soil in the presence of wheat only), it is possible to characterize the effect of weeds on this parameter without taking into account the soil AMF composition. We study the impact of weeds on the number of AMF propagules in soil and not the mechanisms by which weeds impact soil AMF propagules.
Another major comment is “The reported outcomes did not substantiate the mention that weeds have beneficial effects on AMF communities in a winter wheat organic field in Lebanon under authentic field conditions.” I understand that the presence of weeds affected AMF communities, that is the expected results. My question is where are your results that support the beneficial effect on winter wheat organic field?
The beneficial effect of weeds has been demonstrated in the field soil since:
- on the one hand, we have shown that weeds increase the AMF propagule abundance in soils covered by both wheat and weeds compared to those covered only by wheat,
- on the other hand, we have shown that weeds improve the performance of soil in its capacity to enhance plant biomass and root AMF development under pot-controlled experiments. This was done using soil covered by weeds and wheat in pot-controlled experiments. We have shown that soil covered by weeds and wheat (compared to soil covered by wheat only) enhances the growth and the AMF colonization in the roots of host plant Medicago sativa under controlled conditions.
We have clarified our objectives in the revised version of the article (highlighted in green, lines 75-76 and 83-90)
We look forward to hearing from you soon,
Yours sincerely,
Dr Bello Mouhamadou

Reviewer 4 Report
Comments and Suggestions for Authors
The authors corrected all the reviewer's comments or provided a reasoned response. The description of the methodological part of the work has been expanded. The manuscript has become much easier to understand. The weakest aspect of the study remains the lack of study of yield. The authors plan to conduct these studies in the future. Therefore, if this remark is not considered critical, then the manuscript can be published in this form.
Author Response
Grenoble, December 28, 2023
Dear Editor
We would like to thank the reviewer 4 for their remarks / comments, which have helped to improve the manuscript. We have made changes to clarify our objectives.
Reviewer 4
Comments and Suggestions for Authors
The authors corrected all the reviewer's comments or provided a reasoned response. The description of the methodological part of the work has been expanded. The manuscript has become much easier to understand. The weakest aspect of the study remains the lack of study of yield. The authors plan to conduct these studies in the future. Therefore, if this remark is not considered critical, then the manuscript can be published in this form.
Regarding the study of yield, it is not critical since the main objective of the article is to study the effect of weeds on AM fungal community diversity and composition in the roots of wheat and weeds in real field conditions. We have also studied the effects of weeds on the functional performance of soil. To do this, we studied, on the one hand their impact on the number of soil AMF propagules. On the other hand, we studied their impact on the capacity of soil to enhance plant biomass and root AMF development by measuring the growth and the AMF colonization in the roots of host plant Medicago sativa under controlled conditions.
We have clarified our objectives in the revised version of the article (highlighted in green, lines 75-76 and 83-90)
We look forward to hearing from you soon,
Yours sincerely,
Dr Bello Mouhamadou
